# Adherence to the Mediterranean Diet: Impact of Geographical Location of the Observations

**DOI:** 10.3390/nu14102040

**Published:** 2022-05-13

**Authors:** Elisa Mattavelli, Elena Olmastroni, Daniela Bonofiglio, Alberico L. Catapano, Andrea Baragetti, Paolo Magni

**Affiliations:** 1Dipartimento di Scienze Farmacologiche e Biomolecolari, Università degli Studi di Milano, 20133 Milan, Italy; elisa.mattavelli@unimi.it (E.M.); elena.olmastroni@unimi.it (E.O.); alberico.catapano@unimi.it (A.L.C.); andrea.baragetti@unimi.it (A.B.); 2Department of Pharmacy, Health and Nutritional Sciences, University of Calabria, 87036 Rende, Italy; daniela.bonofiglio@unical.it; 3Centro Sanitario, University of Calabria, 87036 Rende, Italy; 4IRCCS MultiMedica, Sesto San Giovanni, 20099 Milan, Italy

**Keywords:** Mediterranean diet, adherence, awareness, health, geographical location, ethnic, nutrition guidelines, chronic disease

## Abstract

The Mediterranean diet has emerged as a comprehensive lifestyle, including specific foods and meal composition and a set of behavioural and social features. Adherence to the Mediterranean diet has been shown to promote health and reduce the prevalence of chronic diseases. The actual implementation of the Mediterranean diet is affected by several sociocultural factors as well as geographical components. Indeed, the geographical location, such as a specific country or different areas in a country and specific latitude and climate, appears to be an important factor that may strongly affect the implementation of the Mediterranean diet or some of its principles as well as the adherence to it. Another dynamic component affecting personal nutritional choices, also regarding adherence to the Mediterranean diet and its principles, is the individual life-long trajectory of food preference and nutrition habits and awareness. In this review, we discuss the current evidence on the impact of geographical location on adherence to the Mediterranean diet.

## 1. Introduction

Among the different nutritional approaches aimed at promoting and maintaining good health as well as preventing chronic diseases, the Mediterranean diet has emerged over the decades as a comprehensive lifestyle, that, in addition to specific foods [1] and meal composition [2,3], also includes a set of behavioural and social features such as the production of specific foods as well as social exchange and communication [4,5,6,7]. The Mediterranean diet reflects the food patterns typical of southern regions of Greece, Italy, Spain and France in the early 1960s, where adult life expectancy was quite high, and the prevalence of diet-related chronic diseases was low [4,5,6]. Such food patterns, based on fresh or minimally processed or refined foods, were more specifically present in rural areas with easier availability of such dietary components, compared to urban areas [8]. The nutritional characteristics of the Mediterranean diet have inspired institutions and experts in several regions of the world and, more specifically, in individual countries, to implement healthier approaches to food consumption into their specific nutritional guidelines [9]. Some examples of this approach include the New Nordic Diet [10] and the US Dietary Guidelines for Americans, 2020–2025, 9th ed. [11]. In this context, it is important to emphasize that, in order to be most effective, nutritional guidelines need to be developed according to several specific factors, such as local cultural, ethnic and traditional features, and geographic and socioeconomic components, as well as to the availability of specific local food products [9]. Among these features, the geographical location (i.e., a specific country, different areas in a country, such as seaside or mountainside or plains far from both, and specific latitude and climate, including climate changes, as well as the related social features) appears to be a crucial factor that may affect to a large extent the actual applicability of the Mediterranean diet or some of its principles as well as the actual adherence to it [12,13,14,15,16,17,18]. An additional dynamic component affecting personal nutritional choices, also regarding adherence to the Mediterranean diet and its principles, is the individual life-long trajectory of food preference and nutrition habits and awareness, including feeding behavior aspects [19]. In the past, this aspect has been associated with local traditions and ethnic features and therefore remained rather stable over time within a specific population. However, over the last decades, due to the exponential diffusion of multiple communication media, the modifications of such traditional nutrition habits have been more frequent, leading to either a nutritional transition to unhealthy habits, especially in younger subjects [17,20,21,22] and in selected areas such as the Middle Eastern and North African (MENA) region [12], or a greater awareness of the health advantages of a good adherence to the Mediterranean diet, especially in middle/older age subjects, with improved nutrition patterns during their lifetime [23]. Thus, the epidemiological and experimental observations reporting adherence to the Mediterranean diet and its impact on health issues should also take into consideration another component: the time-related longitudinal changes within a specific population.

This narrative review discusses the current evidence on the potential impact of these two important factors, geographical location and time-related life-long change, on adherence to the Mediterranean diet. To this aim, the PubMed and Excerpta Medica Database (Embase) were searched from inception until April 2022. The search was also extended to the gray literature. Used search terms, with a combination of MeSH terms if applicable in each database, included: “Mediterranean diet”, “geographical location”, “region”, “regional”, “adherence”, “Nordic diet”, and “age”.

## 2. Adherence to the Mediterranean Diet and Chronic Disease Prevention

Several epidemiological observations and intervention studies demonstrated that implementation of the Mediterranean diet, either as spontaneous adherence [24,25] or following specific intervention approaches (i.e., Lyon Diet Heart Study [26] and PREDIMED Study [27]), are associated with numerous health benefits. These include a reduced incidence of cardiovascular diseases [26,27,28,29,30], including peripheral artery disease [31], diabetes mellitus [32] and the metabolic syndrome [33] and type 2 diabetes mellitus [34,35]. Moreover, adherence to the Mediterranean diet has been found to show positive effects on cancer incidence [36], possibly thanks to the presence of natural compounds with protective effects [37]. Greater adherence to the Mediterranean diet has also been associated with a reduced incidence of neurodegenerative diseases (Parkinson’s and Alzheimer’s disease) [38,39] and, coherently, of cognitive dysfunction and physical impairment [40,41,42].

## 3. Adherence to the Mediterranean Diet and Geographic Location in Adults

As an extension of the above-reported observations, it is important to associate the term “Mediterranean diet” not only with the food quality and meal composition, but also with a particular way of cooking, eating and more. The Mediterranean diet indeed involves a series of skills, knowledge, symbols, rituals, and traditions related to crops, harvesting, animal husbandry, fishing, conservation, processing, and, more specifically, to the sharing and consumption of food [7]. Eating together is the basis of the cultural identity and continuity of communities throughout the Mediterranean area, representing an important moment of social exchange and communication, which results in the reinforcement of the relationships within family, groups, and community [43]. For these reasons, the Mediterranean diet was inscribed in 2013 in the Representative List of the Intangible Cultural Heritage of Humanity Countries: Greece, Italy, Spain, Morocco, Cyprus, Croatia, and Portugal [7]. Based on these considerations, it appears that the specific geographical location and all the related features strongly impact the implementation of the Mediterranean diet or some of its aspects, on the extent of awareness of its benefits and, ultimately, on the adherence of a population. Interestingly, the relevant change in the society features and the associated globalization that occurred in the last half-century and are still ongoing impacted and still impact the relationship between local habits and adherence to the Mediterranean diet, sometimes in opposite directions. A very recent systematic analysis of adherence to the Mediterranean diet among adults in a large set of Mediterranean countries reported that most available studies (in the 2010–2021 period) are from European Mediterranean countries, with fewer studies from Mediterranean countries in North African and Middle Eastern regions [18]. In general, low or moderate adherence was reported by the different studies, without major sex and age differences.

The analysis of adherence to the Mediterranean diet in cross-sectional studies may be useful to unveil possible differences between populations living in the same country or even district, but in different geographical and socio-cultural conditions, such as, for example, in rural areas and in urban contexts, two quite diverse geographical settings. Table 1 reports the main outcomes of the studies conducted in adults and discussed in this section.

One example of this is offered by the comparison between two epidemiological studies recently conducted in northern Italy and rather far from the Mediterranean Sea. One study involved the PLIC cohort, conducted in the urban area of Milan, Italy, on 2500 adult volunteers [44,45] and the other involved the PLIC Chiesa cohort, conducted in about 800 adult subjects in an isolated village in the Italian Alps, at 1000 m above sea level [46]. These two populations, although living just 150 km apart, represent two socio-cultural heterogenous groups: one living in an urban context and the other being a rather homogeneous community with greater isolation and specific socio-cultural identity, respectively. A preliminary analysis of the adherence to the Mediterranean diet, assessed by the PREDIMED 14-item tool [56] (Table 2), was conducted in subgroups from the two studies, according to a case-control design (age- and sex-matched). It was found that PLIC-Chiesa individuals were less adherent to the Mediterranean diet than the corresponding PLIC subjects (mean PREDIMED score 7.38 vs. 8.22, respectively; *p* < 0.001), consistent with the traditional mountain diet, mostly consisting of food of animal origin. The full study will unveil whether this finding is associated with different health outcomes.

A regional variability of adherence to the Mediterranean diet (evaluated by the MD adherence score, Table 2) has also been reported in Spain, where a recent study conducted in all regions showed that southeastern Spain had the lowest score for adherence to the Mediterranean diet specifically related to low consumption of fish and plant products. A lower adherence score to the Mediterranean diet was also strongly associated with the prevalence of hypertension [47]. Studies on adherence to the Mediterranean diet and health outcomes in urban as well as rural areas have also been conducted in Greece. The ATTICA epidemiological study estimated the level of adherence to the Mediterranean diet in a sample (2749 participants) population of the Athens metropolitan area [48], by using the MedDietScore [57] (Table 2). Higher adherence to the Mediterranean diet was observed in areas with a greater proportion of women and older people with a lower unemployment rate and immigrant population, as well as in locations with more green areas and a higher frequency of supermarkets and street markets. On the contrary, adherence to the traditional Mediterranean diet, evaluated by the MedDietScore, in the population of the Elafonisos island, a small Greek island, was found to be moderate and associated with low physical activity and high prevalence of obesity and traditional risk factors for cardiovascular diseases, suggesting the need for lifestyle improvement programs in rural isolated areas of the Mediterranean basin [49]. Another study, conducted in the Republic of Cyprus, showed that adherence to the Mediterranean diet (assessed by the MedDietScore) in a sample of the general population was greater in males and residents of rural regions compared with females and residents of urban regions [50]. Interestingly, adherence to the Mediterranean diet, once consolidated, appears to be rather robust, even under challenging conditions, such as the recent COVID-19 pandemic. In this regard, studies reported that no major changes in Mediterranean diet adherence occurred in Spanish university students [51] and in a sample of the Italian population [52] during the recent COVID-19 lockdown, a major lifestyle challenge. 

Outside the Mediterranean basin, certain data and experiences related to the Mediterranean diet are interesting. Longevity is a relevant advantage associated with adherence to the Mediterranean diet for a substantial part of life [62,63]. To address this issue in Australia, which is a country geographically and culturally quite far away from the Mediterranean area, a prospective cohort study including both Anglo-Celts and Greek-Australians was conducted in Melbourne, with the goal to evaluate whether adherence to the Mediterranean diet affected the survival of elderly people in this developed non-Mediterranean country. The authors reported that a diet that adheres to the principles of the Mediterranean diet was associated with longer survival among Australians of both Greek and Anglo-Celtic origin [53]. 

In the US, the Mediterranean diet, due to its palatability and its consequent high acceptability, is considered a good opportunity for dietary improvement, for example by increasing consumption of fresh vegetables, fruit, grains, and olive oil since the early 20th century [6]. Interestingly, improved awareness of and, consequently, adherence to the Mediterranean diet style in the US has also been promoted by populations that immigrated to the United States from Greece, Italy and Spain. In agreement with previous versions, the 2020–2025 version of the Dietary Guidelines for Americans [11] currently includes the Healthy Mediterranean-Style Dietary Pattern, which is considered a variation of the Healthy US-Style Dietary Pattern, based on the types and proportions of foods typically consumed by Americans, although in nutrient-dense forms and appropriate amounts. A very recent study conducted a geospatial analysis of Mediterranean diet adherence (assessed by the MD adherence score) in the US [54], collecting data across the US regions and exploring the predictive factors of such adherence among US adults (over 20,000 participants). High adherence was observed in 46.5% of the sample. Higher adherence clusters were mainly located in the western and northeastern coastal areas of the USA, whereas lower Mediterranean diet adherence clusters were observed in south and east-north-central regions. Being older, black, not a current smoker, having a college degree or above, an annual household income ≥USD US 75K, exercising ≥4 times/week and watching TV/video <4 h/d were each associated with higher odds of high adherence. The authors concluded that across the US regions there is a significant geospatial and population disparity in adherence to the Mediterranean diet, possibly leading to the greater prevalence of chronic diseases. In the US, although recommended by the current Dietary Guidelines for Americans, as indicated above, the Mediterranean diet is still poorly associated with its benefits, particularly in the Stroke Belt, a large 11-state region in the southeast part of the country with an unusually high incidence of stroke and other forms of cardiovascular disease. A recent study examined Mediterranean diet adherence (assessed by the PREDIMED score) and perceived knowledge, benefits, and barriers in the US [55]. Convenience, sensory factors and health were greater barriers to the Mediterranean diet in the Stroke Belt group, but not in the other groups/regions. Participants with a bachelor’s degree or higher showed greater adherence to the Mediterranean diet, whereas obese participants had a lower adherence. 

When considering the geographical location and Mediterranean diet, it is worthwhile to mention an important European experience, represented by the New Nordic Diet, developed in some northern Europe countries (Denmark, Sweden and Finland), characterized by a markedly colder climate. Here, the plant-based nutrition present in the Mediterranean diet is translated into the consumption of healthy regional-specific foods, such as vegetables available in that area (pears, apples, berries, root and cruciferous vegetables, cabbages, rye bread and whole grain) as well as potatoes, a high intake of fish, low-fat dairy products, and vegetable fats, among other dietary lipid sources [10,64,65]. Moreover, it contains 35% less meat than the average Danish diet and appears to be effective in sustainability terms [66]. Rather little research has been conducted so far on the long-term health effects of adherence to the New Nordic Diet. A recent study found a non-significant inverse association with the overall incidence of myocardial infarction and of stroke in men [67], whereas no association was observed with the incidence of type 2 diabetes mellitus. These findings suggest the need for further studies to highlight which components of the Mediterranean diet provide the well-established cardioprotective effect. In this context, one may speculate about the effects of the different fats used for cooking: olive oil in the Mediterranean diet vs. rapeseed oil in the New Nordic diet [68,69].

Figure 1 reports the main geographical features reported for adults and is discussed in this section.

## 4. Adherence to the Mediterranean Diet and Geographic Location in Adolescents

Although several reports are available regarding adherence to the Mediterranean diet in the adult population, little information is available regarding children and adolescents, spontaneously adhering to this nutrition pattern early in life. Table 3 reports the main outcomes of the studies conducted in adolescents and discussed in this section.

In the context of younger age, education and awareness about the Mediterranean diet and its principles are very important, as shown by a study conducted in Greece and showing that Greek adolescents consume a more westernized diet, which is quite detached from the traditional Mediterranean diet, although actual knowledge about the Mediterranean diet was associated with greater adherence to it [70], according to the evaluation by KIDMED scoring [58] (Table 2). Data from different Italian regions [71] (obtained using the Mediterranean diet scale (MDS) according to [59], revised by [60], Table 2) in a sample of 565 adolescents aged 12–19 years indicate that 38.6% of subjects had low adherence to the Mediterranean diet, whereas only 14% had high adherence. In this study, adolescents from the southern region of Italy showed the highest adherence, compared with those from the northeast and northwest Italy, which are mostly far from the Mediterranean Sea. Recently, in the DIMENU cross-sectional study, carried out in adolescents from the southern Italy area, a medium adherence to the Mediterranean diet assessed by the KIDMED score was reported (medium adherence 60.87%) with no significant differences according to gender [72]. Similar data were also reported by other authors among adolescents living in the Mediterranean region [73], and in the adult population from the same Mediterranean area in which a direct association with age was found [76]. Specifically, Caparello et al. reported that in adulthood the percentage of adherers to recommendations for fruit, nuts, and fish, estimated by the PREDIMED score, was below the dietary guidelines [76]. Supporting this concept, in adolescence, higher consumers of nuts, olive oil and fish showed a better serum metabolic profile, underlining the need to improve the consumption of distinctive components of the Mediterranean diet pattern and to encourage people to change their eating behaviour [77]. In addition, optimal Mediterranean diet adherence (assessed by KIDMED) in adolescents performing vigorous intensity levels of physical activity was found to be associated with lower lipid profile markers and reduced insulin concentrations, reinforcing the healthy benefits of the Mediterranean-style diet pattern [72]. Moreover, serum from adolescents who follow an optimal adherence to the Mediterranean diet (KIDMED) displayed anti-inflammatory and antioxidant properties which may provide chances for the prevention of metabolic and chronic diseases in adulthood [74]. However, a better knowledge of the composition of the healthy Mediterranean diet pattern, in terms of food sources of macro- and micronutrients through nutrition education programs is necessary to improve the adherence to the Mediterranean diet (KIDMED) in the young population. Indeed, the awareness towards the promotion of the Mediterranean diet as a global nutritionally balanced and healthy dietary pattern led to a decreased inflammatory status in adolescents of southern Italy [75].

An interesting “generation shift” from a Mediterranean diet pattern to a more “westernized” type of diet has also been observed in Dalmatia, the Mediterranean coastal part of Croatia, highlighting that younger women, although having a higher education and socioeconomic status, showed a lower adherence to the Mediterranean diet (assessed by the Mediterranean Diet Serving Score [61], Table 2) and a healthier lifestyle than older women from the same region [17]. 

## 5. Conclusions

This paper highlights the relevance of the geographical location and related social features in the actual adherence to the Mediterranean diet, thereby promoting (or not) several health advantages. Moreover, evidence is present that improved health and longevity results from prolonged exposure to this dietary pattern, emphasizing the time factor and the actual moment of its acquisition. However, in several instances, Mediterranean populations have been showing moderate adherence to the Mediterranean diet in the past 10 years [18], which suggests the need for improving adherence to the Mediterranean diet even in the countries of its origin. In the future, the ways in which a palatable and healthful dietary pattern, such as the Mediterranean diet, can be communicated and promoted need to be investigated beyond medical and nutrition authorities, for example, by employing culinary and marketing strategies. Importantly, several critical issues are present when adapting the Mediterranean diet to non-Mediterranean populations [78]. The approach to studying diet–health relationships has progressively moved from individual dietary components to overall dietary patterns that affect the interaction and balance of personal metabolome (low-molecular-weight metabolites) and microbiome (host-enteric microbial ecology). Future studies will be needed to address this complexity, unveiling how metabolome and microbiome profiles are modulated by high adherence to the Mediterranean diet in the different populations and regions to promote health and longevity [79,80].

## Figures and Tables

**Figure 1 nutrients-14-02040-f001:**
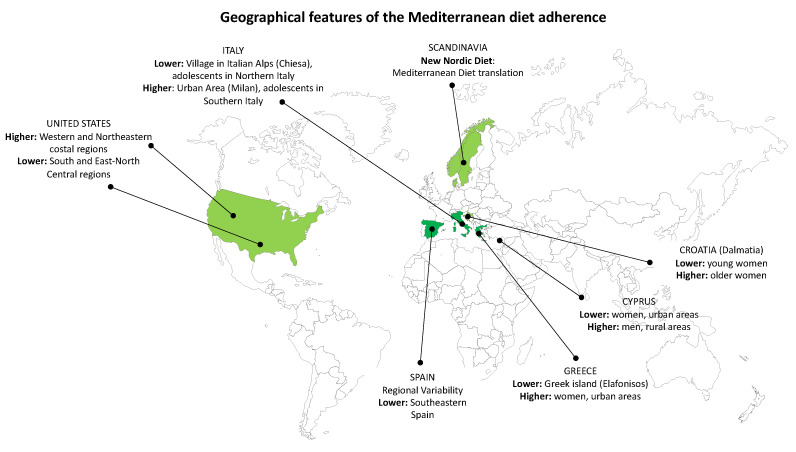
Geographical features of Mediterranean diet adherence in adults.

**Table 1 nutrients-14-02040-t001:** Adherence to the Mediterranean diet and geographic location in adults.

Study Name	Country	Year	Scoring System	Adherence to Mediterranean Diet	Reference
PLIC	Italy (urban)	2017	PREDIMED	higher than PLIC-Chiesa	[44,45]
PLIC-Chiesa	Italy (mountain)	2020	PREDIMED	lower than PLIC	[46]
DIMERICA	Spain (different regions)	2016	MD adherence score	lowest in Southestern Spain	[47]
ATTICA	Greece (urban)	2021	MedDietScore	higher in women, older people	[48]
PERSEAS	Greece (island)	2017	MedDietScore	moderate, despite being in a small island	[49]
	Cyprus	2021	MedDietScore	higher in males/rural areas	[50]
	Spain	2022	PREDIMED	no major changes during COVID-19 lockdown	[51]
	Italy	2020	PREDIMED	no major changes during COVID-19 lockdown	[52]
	Australia	1999	composite score	higher was associated with longer survival	[53]
	USA (different regions)	2020	MD adherence score	significant geospatial/population disparities	[54]
	USA (Stroke Belt/other regions)	2019	PREDIMED	lower in Stroke Belt group	[55]

**Table 2 nutrients-14-02040-t002:** Mediterranean diet score systems used in the reported studies.

Scoring System	Range for Classification in Categories		
	Low	Moderate	High
PREDIMED (MEDAS) [56]	≦5	6–9	≧10
MD adherence score [47]	<5	5–7	>7
MedDietScore [57]	<25/55	26–35/55	>35/55
KIDMED [58]	≦3	4–7	8
Mediterranean diet scale (MDS) [59], [60]	0–3	4–5	6–9
Mediterranean Diet Serving Score (MDSS) [61]	0–9.9	10–13.9	≧14 (24 max)

Mediterranean diet scores were assessed by direct administration by a trained dietician (PREDIMED) or by evaluating a food frequency questionnaire (all others). The related references are indicated.

**Table 3 nutrients-14-02040-t003:** Adherence to the Mediterranean diet and geographic location in adolescents.

Study Name	Country	Year	Scoring System	Adherence to Mediterranean Diet	Reference
	Greece	2009	KIDMED	greater if good knowledge of Med diet	[70]
MDFS group	Italy	2014	Mediterr. diet scale	low: 38.6%; high: 14%. Higher in southern Italy	[71]
DIMENU	Italy (Southern)	2020	KIDMED	medium: 60.9%	[72]
	Italy (Sicily)	2013	KIDMED	moderate in general, lower in urban settings	[73]
	Italy	2021	KIDMED	higher associates to antiinflammatory profile	[74]
DIMENU	Italy (Southern)	2021	KIDMED	higher with education programs	[75]
	Croatia (Dalmatia)	2021	MDSS	reduced in younger women	[61]

## Data Availability

Not applicable.

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
