# Peer review of "Adherence to the Mediterranean Diet: Impact of Geographical Location of the Observations"

_nutrients, 2022, doi:10.3390/nu14102040_

Round 1
Reviewer 1 Report
This is a good paper to understand the relevance of all the characteristics that make up the Mediterranean diet.
In line 35 maybe you can include more countries like Spain or south of France, they also are Mediterranean countries, or with med diet
In line 37 you can explain main differences in dieta between people who lives in cities or rural areas. i
in line 176 you talk about Northern Europe countries and their change about Mediterranean diet but benefits don’t appear. Could be the difference of fat they have used to cook?
Author Response
REVIEWER 1
Comments and Suggestions for Authors
This is a good paper to understand the relevance of all the characteristics that make up the Mediterranean diet.
Thank you for your positive comment.
In line 35 maybe you can include more countries like Spain or south of France, they also are Mediterranean countries, or with med diet
Thank you, done. Croatia, Cyprus and the Middle Eastern and North African (MENA) region have been added.
In line 37 you can explain main differences in diet between people who lives in cities or rural areas.
Thanks, this comment has been added in line 39.
in line 176 you talk about Northern Europe countries and their change about Mediterranean diet but benefits don’t appear. Could be the difference of fat they have used to cook?
Thank you for this comment. This observation has been added and referenced in lines 197-199.
Reviewer 2 Report
Dear Authors,
The Mediterranean diet has aroused much interest among researchers, nutritionists and ordinary people because of its positive relationship with health. Hence, I find the choice of topic appropriate and challenging.
After reading the article, I find that many issues are presented in a very general way. Thus, there is a lack of detailed focus on the issues contained in the title of the paper, i.e. awareness.
The purpose of the article should be clarified. The discussion should be more focused. The relationship between variables was discussed, not the effect, and at least cross-sectional studies cited provide such opportunities. The conclusions in the final section of the article do not reflect the issues discussed.
The article is based on the literature, but its use is questionable due to its small number. For example, a large section of the Introduction (lines 46-58) in which the choice of two factors as relevant to the adherence to the Mediterranean diet is justified does not appear any references. Although "awareness" appears in the title, it is not sufficiently reflected in the paper. It may be worth limiting consideration to adherence to the diet only.
In my opinion, the discussion of the current evidence about the impact of geographical location and time-related life-long change, "of awareness and adherence to the Mediterranean diet" (lines 62-64) is insufficient. Moreover, up to this point , it has not been substantiated and therefore it is rather unclear what its purpose is. Too few studies cited in the paper.
Presenting differences due to geographic location would be easier if specific geographic indicators, such as distance from the coastline, location above sea level, etc., were included in the geographic characteristics. On the other hand, focusing only on describing the results of research conducted in different countries is not the best solution. The understanding of Figure 1. „Geographical features of the Mediterranean diet adherence” is difficult, but it indicates that only a few countries were included in the discussion.
Each literature review is performed according to a specific methodology, which includes information on how to search for sources, but also defines variables. This work lacks methodological assumptions.
The section „Adherence to the Mediterranean diet: impact of time and related variables” needs refinement due to poor literature coverage.
Such conclusion as „Further research aimed to define the mechanisms of the Mediterranean diet in protecting against chronic diseases, possibly by promoting specific feeding behaviors [45], is required” is not legitimate in light of the work presented.
In my opinion the manuscript requires further work primarily to take into account other available studies, but also to refine the purpose and methodology.
Author Response
REVIEWER 2
Comments and Suggestions for Authors
Dear Authors,
The Mediterranean diet has aroused much interest among researchers, nutritionists and ordinary people because of its positive relationship with health. Hence, I find the choice of topic appropriate and challenging.
After reading the article, I find that many issues are presented in a very general way. Thus, there is a lack of detailed focus on the issues contained in the title of the paper, i.e. awareness.
The purpose of the article should be clarified. The discussion should be more focused. The relationship between variables was discussed, not the effect, and at least cross-sectional studies cited provide such opportunities. The conclusions in the final section of the article do not reflect the issues discussed.
Thanks, all sections have been revised/integrated according to your suggestions, and especially better focusing on the issues that you identified.
The article is based on the literature, but its use is questionable due to its small number. For example, a large section of the Introduction (lines 46-58) in which the choice of two factors as relevant to the adherence to the Mediterranean diet is justified does not appear any references. Thanks, 24 additional appropriately selected references have been added. Eleven relevant references have been added to the section at lines 46-58, as suggested.
Although "awareness" appears in the title, it is not sufficiently reflected in the paper. It may be worth limiting consideration to adherence to the diet only.
Thanks, title, abstract and text have been modified accordingly.
In my opinion, the discussion of the current evidence about the impact of geographical location and time-related life-long change, "of awareness and adherence to the Mediterranean diet" (lines 62-64) is insufficient.
Thank you, the term “awareness” and the related issues have been removed from the paper, as indicated above. The paper has been estensively revised and further references have been added, a indicated above.
Moreover, up to this point , it has not been substantiated and therefore it is rather unclear what its purpose is. Too few studies cited in the paper.
Thanks, we added 24 new references to the paper. As mentioned above, any reference to “awareness” has been deleted.
Presenting differences due to geographic location would be easier if specific geographic indicators, such as distance from the coastline, location above sea level, etc., were included in the geographic characteristics.
Thanks, some geographic indicators have been added, when available in the original papers.
On the other hand, focusing only on describing the results of research conducted in different countries is not the best solution. The understanding of Figure 1. „Geographical features of the Mediterranean diet adherence” is difficult, but it indicates that only a few countries were included in the discussion.
Thanks, Figure 1 has been revised and more information have been implemented.
added countries
Each literature review is performed according to a specific methodology, which includes information on how to search for sources, but also defines variables. This work lacks methodological assumptions.
Thanks, a methodological section has been added at the end of section 1. (Introduction).
The section „Adherence to the Mediterranean diet: impact of time and related variables” needs refinement due to poor literature coverage.
Thanks, we added appropriate references and revised the text.
Such conclusion as „Further research aimed to define the mechanisms of the Mediterranean diet in protecting against chronic diseases, possibly by promoting specific feeding behaviors [45], is required” is not legitimate in light of the work presented.
Thanks, the sentence has been deleted.
In my opinion the manuscript requires further work primarily to take into account other available studies, but also to refine the purpose and methodology.
Thank you for all these constructive comments. The paper has been revised according to other available studies, and purpose and methodology have been refined/added.
Reviewer 3 Report
This is a very interesting review paper in which authors try to discuss the current evidence about the impact of geographical location and time-related life-long change of awareness and adherence to the Mediterranean diet. However it needs some major revisions:
Introduction section, p.2: From line 46 to line 58 there is no reference. However, some need to be added.
p.2, line 79: please remove the second "not".
Overall: I believe that there are many shortcomings in this narrative review paper. First of all, there is no clear review methodology. The figure cannot be considered correct because many articles have been omitted, and therefore the conclusions are biased. For example, in the case of Greece, many studies have been done on eating habits and, more specifically, on the level of adherence to the Mediterranean diet; however, they are never mentioned in the article. Moreover, in the figure, the notes do not have a common logic of existence. For example, in the case of Italy, Greece and the USA, there is a footnote "Lower" and "Higher", in Spain only "Lower" and in the Scandinavian countries "New Nordic Diet". In other words, there is no specific information pattern. I also think that the other Mediterranean countries in the wider Mediterranean basin should be mentioned....
Author Response
REVIEWER 3
Comments and Suggestions for Authors
This is a very interesting review paper in which authors try to discuss the current evidence about the impact of geographical location and time-related life-long change of awareness and adherence to the Mediterranean diet.
Thanks for this positive comments.
However it needs some major revisions:
Introduction section, p.2: From line 46 to line 58 there is no reference. However, some need to be added.
Thanks, 11 references have been added in the indicated part.
p.2, line 79: please remove the second "not".
Thanks, done.
Overall: I believe that there are many shortcomings in this narrative review paper. First of all, there is no clear review methodology.
Thanks, a methodology paragraph has been added at the end of the Introduction section.
The figure cannot be considered correct because many articles have been omitted, and therefore the conclusions are biased. For example, in the case of Greece, many studies have been done on eating habits and, more specifically, on the level of adherence to the Mediterranean diet; however, they are never mentioned in the article.
Moreover, in the figure, the notes do not have a common logic of existence. For example, in the case of Italy, Greece and the USA, there is a footnote "Lower" and "Higher", in Spain only "Lower" and in the Scandinavian countries "New Nordic Diet". In other words, there is no specific information pattern.
Thanks, the figure has been updated and modified accordingly.
I also think that the other Mediterranean countries in the wider Mediterranean basin should be mentioned....
Thanks, we fully agree. Croatia, Cyprus and the Middle Eastern and North African (MENA) region have been added.
Round 2
Reviewer 2 Report
Thank you, I accept the changes.
Author Response
Thank you for your valuable comments.
Reviewer 3 Report
The revised version of the manuscript entitled "Awareness and Adherence to the Mediterranean Diet: Impact of Geographical Location and Time-frame of the Observations" is better than the previous version.
However I still believe that the figure is not very well comprehensive or justified and maybe it should be removed.
Author Response
Thank you for your valuable comments.